# Saliva Secretion and Swallowing—The Impact of Different Types of Food and Drink on Subsequent Intake

**DOI:** 10.3390/nu12010256

**Published:** 2020-01-19

**Authors:** Catherina Bozorgi, Celina Holleufer, Karin Wendin

**Affiliations:** 1Faculty of Health Sciences, University of Southern Denmark, DK-5230 Odense, Denmark; catbz@outlook.com (C.B.); celina_holleufer@outlook.dk (C.H.); 2Faculty of Natural Sciences, Kristianstad University, SE-291 39 Kristianstad, Sweden; 3Department of Food Science, University of Copenhagen, DK-1958 Frederiksberg C, Denmark

**Keywords:** food oral processing, nutrition, malnutrition, ease of swallow

## Abstract

The oral processing of food is important for eating and digestion in order to gain energy and nutrients. Due to disease, injury, or aging, individuals may experience difficulties in this process. These difficulties often lead to dysphagia, which is associated with malnutrition. Thus, it is of importance to find solutions and strategies to enable food intake. It is well known that sour and/or carbonated foods and drinks increase saliva secretion and trigger the swallowing reflex. However, knowledge regarding how subsequent food intake is impacted is lacking. The aim of this study was to clarify whether sour and/or carbonated foods and drinks had subsequent impacts on swallowing function. Twelve healthy participants evaluated eleven foods and drinks in terms of their ability to increase saliva production and ease the swallowing of subsequent food. Results showed that sourness and carbonation had positive impacts on saliva secretion and swallowing. No correlation was found between the pH/sourness of the foods and the ease of swallowing them. It was concluded that the ingestion of cherry tomatoes, natural yoghurt, and, in particular, citrus juice made swallowing of a neutral cracker easier. These results may be used to increase food intake among dysphagia patients.

## 1. Introduction

The oral processing of food is extremely important for the ability to consume and digest foods in order to gain necessary energy and nutrients, as well as to derive pleasure from the recognition and appreciation of food flavors. A review of knowledge regarding food oral processing was presented by Jianshe Chen in the article “Food oral processing: Mechanisms and implications of food oral destruction” [1]. This review described background knowledge of the oral mechanism, including the structural breakdown of food once it enters the mouth and how different control mechanisms optimize the processes of breakdown and perception of foods and the eating experience [1]. Multiple oral operations occur when foods are consumed, such as biting, chewing and mastication, transportation, bolus formation, and swallowing [1,2]. Many people experience difficulties in conducting these oral operational steps due to disease, injury, or complications related to aging. For some individuals, these problems can lead to dysphagia [3]. Aging commonly causes muscle strength weakening of the tongue, thereby leading to a reduced ability to masticate [4]. The tongue plays a particular role in food digestion; it is a mechanical device that helps break food into particles whilst mixing it with saliva to form a bolus [4]. Dysphagia is, therefore, a well-known problem, especially among the elderly.

According to measurements by the European Society for Swallowing Disorders (ESSD), the prevalence of dysphagia is estimated to be between 30% and 40% in independent older people, 44% in those admitted to geriatric acute care, and 60% in institutionalized older people [5]. The ESSD defines dysphagia as a major problem due to its association with malnutrition [6]. Dysphagia and malnutrition are conditions that frequently appear together in hospitalized patients, especially those in long-term care [7]. Malnutrition as a result of suboptimal food intake, which eventually leads to a lack of macro- and micronutrients, has major consequences for health [8].

Many elderly individuals develop dysphagia due to age-related changes in the mouth and the pharyngeal muscles [8]. They are therefore particularly at risk, since dysphagia most often leads to reduced intake of energy and nutrients which, together with lower activity levels, may have serious consequences for the body’s functionality. These consequences can be characterized as malnutrition and a decrease in muscle mass, otherwise referred to as sarcopenia, which can lead to frailty and weakened bone structure [9]. To avoid such consequences, individuals with dysphagia need to be given strategies to enable them to consume sufficient food and fulfil their nutritional and energy requirements. The inability to ingest regular food and drink can also have social impacts for elderly individuals with dysphagia, due to the shame associated with not being able to participate in social meals and the psychological fear of choking during food ingestion. For many with dysphagia, these factors can lead to a reduced quality of life as a result of social isolation [10]. Although texture-modified foods and thickened liquids were developed for individuals with dysphagia, [11] dysphagia patients may have difficulty identifying and consuming enough food with textures and consistencies that they find easy to swallow. Consequently, international levels of texture-modified diet and thickened drinks, as defined by International Dysphagia Diet Standardisation Initiative (IDDSI), may enable patients to maintain adequate nutritional intakes [12].

The main problems for people experiencing dysphagia are due to functional complications: (1) Food or liquid entering the air passage (aspiration) and (2) the possibility of food accumulating in the throat after eating [7]. These complications and discomforts can also affect the whole meal experience and, when eating the next meal, the feeling of pleasure may then be replaced with aversion and fear [10]. These complications illustrate how important it is to find solutions for dysphagia patients to enable them to increase food intake and nutritional status and relieve suffering during chewing and swallowing.

It is well known that foods and drinks that are sour and/or carbonated increase saliva secretion, thereby triggering the swallowing reflex [13,14]; however, these effects are impacted by the carrier (foods/drinks) of the sourness/carbonation. Elshukri et al. (2015) showed how saliva production and swallowing were affected by different carriers [13]. Miura concluded that both sour and carbonated products affected swallowing function [15]. Both studies were used to compare and relate our data results to similar research. Therefore, the aim of this project was to determine if sour foods and carbonated drinks had subsequent impacts on swallowing function.

The hypothesis was that sourness and carbonation would manipulate the signal for secretory events and swallowing differently, thereby leading to a discussion regarding the following three research questions: Which test foods are the easiest to swallow? Does the level of pH/sourness correlate with ease of swallowing the test food? Does the test food affect the subsequent consumption of foods and drinks and is there a causal coherence?

## 2. Materials and Methods

### 2.1. Participants

The test was conducted in a healthy study population and was characterized as a pilot study. The criteria for subject selection included people aged 18 years or older and with no dysphagia diagnosis. The participants consisted of 12 healthy subjects aged 21–54 years old (mean = 33), of whom 2 were male.

Before the test, the subjects were briefed on the aim of the study and given instructions on how to complete the test. They were informed that their participation was voluntary and that they were free to leave the test whenever they wanted. All subjects gave their informed consent for inclusion before they participated in the study. Each test session lasted approximately thirty minutes and no longer than fifty minutes. Warm beverages and sweets were offered after the test.

### 2.2. Preparation of Food Samples

The following eleven foods and drinks were used in the test: Citrus juice (ICA, Sweden), sparkling water (Loka, Sweden), sparkling water with raspberry and blueberry (Loka), natural yoghurt (Valio, Finland), mango sorbet (SIA, Sweden), cornichons (Felix, Sweden), grapefruit, apples, cherry tomatoes, sourdough bread (Pågen LantGoda Surdegsbröd, Sweden), and sodium bicarbonate (Samarin, Sweden) dissolved in water. Sodium bicarbonate is traditionally used to treat heartburn. All of the fruits were bought as single items and came from Sweden.

The foods and drinks were all served on a tray in 40 mL plastic cups, with one sample of each food/drink for every participant. All test samples were prepared on the day of the test and used when fresh. Between the test foods/drinks, the participants were asked to neutralize their palate with a plain cracker (Smörgåsrån, Göteborgs Kex, Sweden).

### 2.3. Execution of the Test

The subjects sat in individual booths in a sensory lab equipped according to International Organization for Standardization (ISO) standard 8589:2010. Before beginning the test, they completed an initial questionnaire on demographics. Basic data were collected prior to evaluation of the products, i.e., initial amounts of saliva and ease of swallowing. The subjects then received one questionnaire per food sample yielding a further 11 questionnaires, which were all identical. The questionnaires consisted of 3 pages each (Figure 1). The food samples were served in random order. The subjects answered the questionnaire for each product they consumed by making a mark on a scale measuring 12 cm in length.

From questions 1a and 1b (Figure 1), we created a baseline by asking the participants to determine the perceived amount of saliva in their mouths before tasting a cracker and how easy it was to swallow the cracker. In addition, pH was measured in duplicate using a pH-meter (Mettler Toledo, EL20, Columbus, OH, USA).

### 2.4. Data Processing

The data were evaluated in Microsoft Excel Version X, version 16.21.1 and mean values and standard deviations were calculated. T-tests were performed on the resulting data to identify significant differences between samples. In order to compare the resulting data, baselines for the perceived amount of saliva and ease of swallowing (questions 1a and 1b) were set to zero. The initial baseline values were then subtracted from the results of the measurements to enable comparisons between the test foods. Pearson’s correlation was used to evaluate whether there was a correlation between the saliva before tasting the test food and the ease of swallowing the cracker after each test food. The purpose was to check if there was an order effect and if the measurements were independent.

## 3. Results

The aim of this study was to clarify whether sour and/or carbonated foods and drinks had subsequent impacts on swallowing function. The results are presented in this section and are discussed in relation to previous studies in the following section.

Table 1 provides an overview of all the tested foods and drinks and the effect they had on perceived sourness, saliva level, and swallowing.

### 3.1. Test Food—Ease of Swallowing and Perceived Amount of Saliva

The results (Table 1) showed that all of the test foods were easier to swallow than the test cracker, except for cornichons and sourdough bread, where the values were negative compared to the baseline. All of the test foods, except for the two previously mentioned and cherry tomatoes, were significantly (*p* ≤ 0.05) easier to swallow than the test cracker (Figure 2).

When comparing the test foods with each other, sparkling water, yoghurt, and citrus juice were the easiest to swallow. These were significantly easier to swallow than most of the other test foods (Table 1).

The perceived amount of saliva compared to the baseline cracker is shown in Figure 3. All of the test foods, except for sourdough bread, apples, and mango sorbet, increased the perceived amount of saliva. However, as = seen in Figure 3, only three of the test foods, namely, citrus juice, cherry tomatoes, and grapefruit, increased the perceived amount of saliva significantly (*p* ≤ 0.05). When comparing the test foods to each other, citrus juice increased the perceived amount of saliva significantly (*p* ≤ 0.05) more than all the other test foods (Table 1).

The perceived amount of saliva did not correlate with ease of swallowing.

The perceived sourness was significantly (*p* ≤ 0.05) higher for citrus juice compared to all the other test foods and was the lowest for the sourdough bread (Table 1). The perceived sourness was also high for cornichons, yoghurt, and grapefruit. Perceived sourness did not correlate with the pH of the test foods. The test results showed no tendency for pH and sourness to correlate with ease of swallowing or the perceived amount of saliva. However, citrus juice and sourdough bread appeared at opposite ends of the scale for all of the measurements.

### 3.2. Ease of Subsequent Swallowing of Neutral Food

As indicated in Table 1, citrus juice made it easier to swallow the cracker. Yoghurt, cherry tomatoes, and mango sorbet also made it easier to swallow a neutral food such as a cracker. These foods were significantly (*p* ≤ 0.05) more effective than the other test foods in this respect. However, compared to the baseline, only citrus juice had a statistically significant impact. The test results showed that sourdough bread, sodium bicarbonate, and (neutral) sparkling water did not ease the participants’ subsequent swallowing of the cracker.

No correlations were found between the ease of subsequent swallowing of the cracker and the other measured parameters, i.e., the perceived sourness, the amount of saliva, and the ease of swallowing the test food.

### 3.3. Correlations

To ensure that the order effect did not impact the results, calculations were performed to check that no correlations occurred between saliva before tasting the test food and ease of swallowing the cracker after each test food.

As mentioned, the results from the correlation calculations indicated that no significant correlation was found between the perceived amount of saliva before intake of the test foods and after, and the ease of swallowing a cracker before and after the test foods.

## 4. Discussion

The finding that citrus juice eased the swallowing of food ingested after its intake was in agreement with a study by Mulheren (2016), which showed that sour products increased swallowing frequency after intake [16]. This was also seen in our results, where there was an increase in the ease of swallowing the cracker after most of the test products. More specifically, the Mulheren study revealed that sour products had an increasing effect on the hemodynamic responses, where motion occurred 0–2 s after bolus onset and when the last step in swallowing occurred.

Our results indicated that sourdough bread did not ease the swallowing of the food ingested afterward, however, citrus juice, cherry tomatoes, and yoghurt did. This result matched other findings for citrus juice, mainly the mean value effect that citrus juice had on easing swallowing after its intake.

This might have been caused by a combination of the viscosity, smoothness, and acidity of the test foods. It was shown that foods with higher acidity levels and slipperiness, for instance, yoghurt and citrus juice, were easier to swallow since they did not require chewing. In addition, their acidity increased saliva production, which then made swallowing a non-sour food after ingestion of the yoghurt or juice easier [17].

Carbonated drinks are of specific interest. These products have pH values between 5.43–5.73, which is markedly under the neutral 7, but higher than most sour products (e.g., citrus juice). At the same time, carbonated drinks are very easy to swallow, therefore, good results could be obtained due to the combination of sourness with carbonation.

What we found was interesting, not least since supporting evidence was observed by Elshukri et al. (2015) [13]. The Elshukri study showed that the combination of carbonation and citric acid, or sour products, had a potentially beneficial impact on swallowing. More specifically, the study clarified that sensorial stimulation increases with carbonation, sour products, and citric acid. This leads to an increased somatosensory input and, together with the carbonation and citric acid, changes in excitation of the pharyngeal corticobulbar tract are provoked [18], meaning that swallowing occurs more easily. In addition, this sensorial combination of carbonation and citric acid creates a bolus consistency that was proven to result in a faster swallowing process [19]. However, in our study, carbonated drinks did not have a significant impact on the subsequent swallowing of the neutral cracker.

These results may be compared to a cohort study where the effect of still water on swallowing was tested and found not to have the same impact on the subsequent bolus consistency and ease of swallowing based on swallowing reaction-time tasks [20]. Some significant similarities to our results were observed, e.g., our study found that the carbonated drinks—sparkling water with raspberry, regular sparkling water, and sodium bicarbonate—were easiest to swallow by themselves. At the same time, data from the cohort study showed that all of the products tested were categorized as sour, as determined by their pH values. In accordance with the results of Miura (2009), the combination of a sour product and a carbonated drink led us to understand that these products had positive impacts on swallowing function [15].

In our results, no order effect was observed regarding the sequence in which the test foods were ingested, thus eliminating this as a factor impacting ease of swallowing.

Although not statistically significant, this study showed that tomatoes had an impact on swallowing the cracker that was eaten afterwards. A study by Krop et al. (2018) specified that tomato soup had an impact on short oral exposure time after intake, potentially supporting our findings that tomatoes tended to trigger saliva secretion [21]. This finding may be an interesting further research direction, first from the physiological aspect of using food components to ease the swallowing of subsequent foods, and second with regard to social aspects through impacting the ability to consume regular foods and gain pleasure from a whole food experience. To support this, a survey by the sociologist Fischler (1988) pointed out that, in food culture, the social aspects and the health-promoting benefits are decisive for human identity with regard to both the nutritional physiological function and the symbolic function of being able to eat the same food as others when eating in communities [22]. Persons with dysphagia can, therefore, experience huge limitations in their everyday life because of food becoming a narrower “source of survival” [23]. In relation to the findings concerning foods/drinks that affected subsequent consumption, physiological positives in preventing malnutrition may be present, as well as symbolic and social positives regarding the ability to eat regular foods.

As seen in the last column of Table 1, three products—sodium bicarbonate, sourdough bread, and sparkling water—stood out negatively in our results and, to some extent, contradicted the hypothesis. These products did not fully confirm that sour products manipulated saliva secretion. So, why was this? The sodium bicarbonate was closer to neutral, with a pH value of 5.73, which, because of its lack of sourness, may indicate why the sodium bicarbonate did not make it easier to swallow the cracker. Even though the sourdough bread had a pH value of 3.58, it contained various ingredients, such as flour, water, eggs, and salt, which could explain the low indication of ease of swallowing the cracker after tasting the sourdough bread. The sparkling water also had a pH value of 5.61 and was therefore closer to neutral, which could be why this product did not make it easier to swallow the cracker. Results from this study showed that sparkling water with raspberry, rather than plain sparkling water, made it easier to swallow a subsequent food. This could be explained by the pH value of sparkling water with raspberry being 5.43, which was slightly lower than the plain sparkling water. The added raspberry, which is a sour food and therefore added extra sourness to the liquid, might have caused the difference between the two different sparkling waters tested. These results confirmed that sour products with a pH closer to neutral had smaller effects on swallowing. Parallel results were also presented in a study by Asselin and Dietsch (2016), which clarified that very sour liquids improved swallowing mechanics in individuals with dysphagia [20]. They showed that the sour products also had tendencies to reduce the amount of excessive food scraps collecting in the throat. More specifically, food accumulates in the vallecular area, which is a major problem for most dysphagia patients due to the increased likelihood of discomfort and coughing. 

Our results specified that carbonated and sour products increased perceived saliva secretion. Even though this study showed that products that were less sour, such as tomatoes and natural yoghurt, were conducive to the swallowing mechanism, the study also revealed that citrus juice had a significantly positive impact on perceived saliva secretion. For future reference, it would be recommended to direct further study toward foods with similar impacts on salivation and swallowing as citrus juice, but with lower acid concentrations. Yoghurt and cherry tomatoes were easy to swallow; they are also only moderately sour and therefore do not compromise oral health [24]. These results could be used in a real context and to direct further research toward developing strategies for dysphagia patients. However, the high acidity of citrus juice would have negative impacts on the teeth, tooth enamel, and oral hygiene in the long term [25]. It is common knowledge that dysphagia often affects the elderly, who often have poor oral health as a result of inadequate oral hygiene and dry mouth [26], which is an important factor since the detection of sour taste is impaired in patients with high growths of oral bacteria [24]. Maintaining adequate oral hygiene in hospitalized elderly persons should therefore be of high priority to avoid reduction of the perception of sour taste and the possible negative influence on swallowing for dysphagia patients [8]. Oral health should therefore be taken into consideration in further studies.

## 5. Conclusions

We conclude that sourness and carbonation can manipulate the signal for secretory events and swallowing. The easiest food to swallow was sparkling water. No correlation was found between pH/sourness and ease of swallowing the test food. Furthermore, we found that there was no order effect and therefore no correlation between the pH value and the order of ingestion of each test food. The results revealed that the ingestion of cherry tomatoes, natural yoghurt, and, in particular citrus, juice made it easier to swallow a neutral cracker.

These overall results could be used in a real context, in interdisciplinary work, and to form strategies to help dysphagia patients maintain an improved lifestyle.

## Figures and Tables

**Figure 1 nutrients-12-00256-f001:**
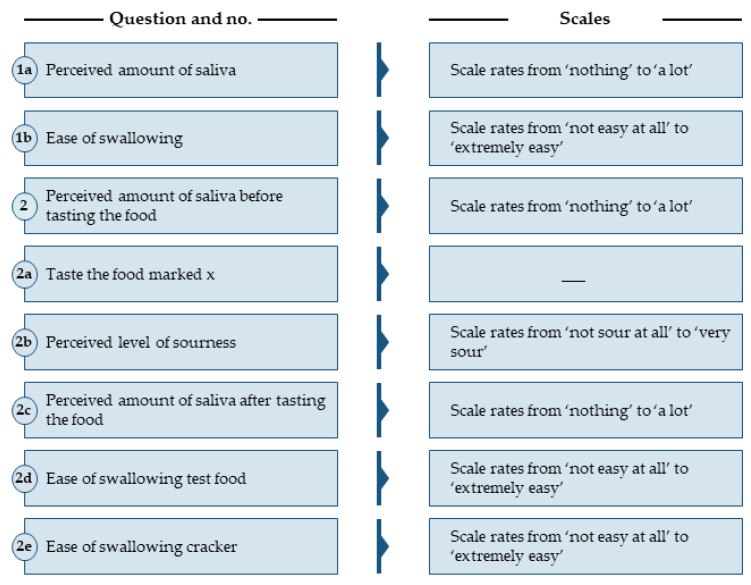
Questions 1a and 1b formed the baseline for questions 2a–2e. These questions were repeated in all eleven questionnaires, i.e., for each product the subjects consumed. The participants made a mark on a scale measuring 12 cm in length.

**Figure 2 nutrients-12-00256-f002:**
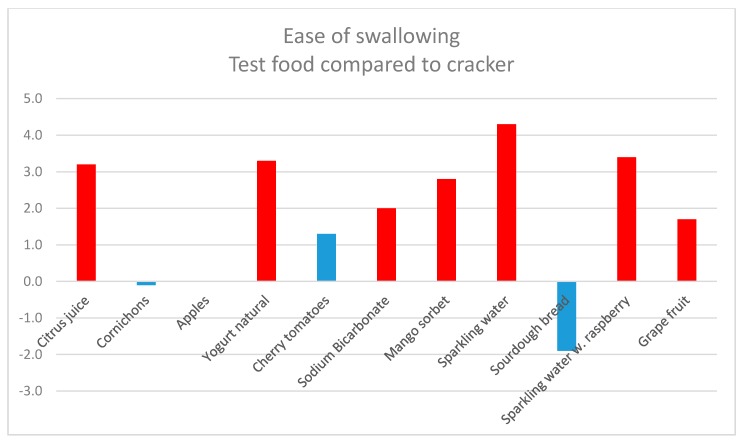
Ease of swallowing test foods compared to the baseline cracker. The red bars indicate a significant difference from the baseline, *p* ≤ 0.05.

**Figure 3 nutrients-12-00256-f003:**
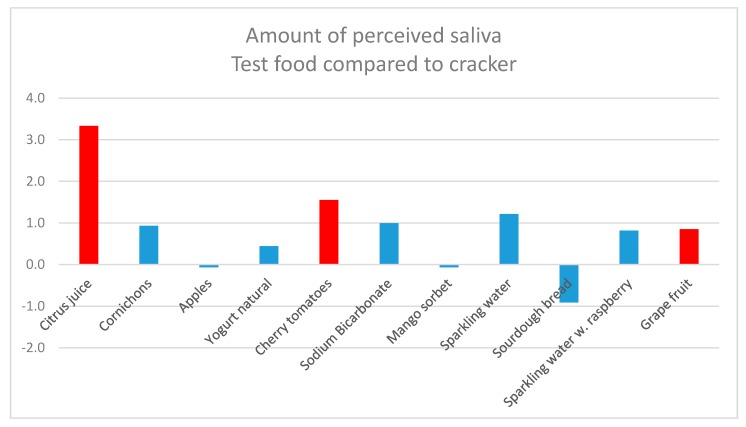
Perceived amount of saliva for test foods compared to the baseline cracker. The red bars indicate a significant difference from the baseline, *p* ≤ 0.05.

**Table 1 nutrients-12-00256-t001:** Results for pH, perceived sourness, saliva level, and ease of swallowing. Different letters in the same column indicate significant differences between samples at *p* ≤ 0.05.

Test Food	pH	Sourness Test Food	Saliva after Test Food *	Ease of Swallowing Test Food *	Ease of Swallowing Cracker after Test Food *
	(m ± sd)	(m ± sd)	(m ± sd)	(m ± sd)	(m ± sd)
**Citrus juice**	2.51 ± 0.04	11.7 ± 1.1 ^a^	3.3 ± 2.3 ^a^	3.2 ± 1.4 ^aef^	2.9 ± 2.8 ^a^
**Cornichons**	3.34 ± 0.04	8.9 ± 1.2 ^cd^	0.9 ± 2.6 ^b^	−0.1 ± 2.4 ^bc^	0.1 ± 2.0 ^b^
**Apples**	3.86 ± 0.05	5.5 ± 1.9 ^b^	−0.1 ± 1.6 ^e^	0.0 ± 2.5 ^bd^	0.3 ± 1.7 ^b^
**Yoghurt, natural**	4.26 ± 0.02	7.9 ± 2.0 ^de^	0.4 ± 2.2 ^bcd^	3.3 ± 1.1 ^af^	0.9 ± 1.8 ^a^
**Cherry tomatoes**	4.48 ± 0.13	6.2 ± 2.6 ^b^	1.6 ± 1.8 ^d^	1.3 ± 2.2 ^cde^	1.4 ± 2.3 ^a^
**Sodium Bicarbonate**	5.78 ± 0.03	7.1 ± 1.6 ^f^	1.0 ± 1.7 ^b^	2.0 ± 3.0 ^acd^	−0.4 ± 1.5 ^b^
**Mango sorbet**	3.57 ± 0.07	5.0 ± 2.7 ^b^	−0.1 ± 2.3 ^bc^	2.8 ± 0.9 ^af^	0.7 ± 2.5 ^a^
**Sparkling water**	5.58 ± 0.02	4.0 ± 2.6 ^b^	1.2 ± 2.2 ^bc^	4.3 ± 1.0 ^a^	−0.3 ± 3.8 ^bd^
**Sourdough bread**	3.71 ± 0.05	1.7 ± 2.2 ^c^	−0.9 ± 2.1 ^c^	−1.9 ± 2.6 ^b^	−1.7 ± 2.1 ^c^
**Sparkling water with raspberry flavour**	5.37 ± 0.04	4.9 ± 3.3 ^b^	0.8 ± 1.8 ^b^	4.4 ± 3.3 ^a^	0.5 ± 4.2 ^bd^
**Grapefruit**	3.58 ± 0.02	7.3 ± 1.9 ^e^	0.9 ± 2.9 ^b^	1.7 ± 2.2 ^cdf^	0.7 ± 2.0 ^b^

* compared to baseline.

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
