# Peer review of "Saliva Secretion and Swallowing—The Impact of Different Types of Food and Drink on Subsequent Intake"

_nutrients, 2020, doi:10.3390/nu12010256_

Round 1

Reviewer 1 Report

the whole article must be submitted to a native English speaker for language revision (introduction and discussion especially). Many terms do not sound very clear (as for example the expression "physiological constellation of food oral processing" is a bit too wordy and confusing) lines 68-69 belong to the conclusion and should not be presented in the introduction; "Samarin" (line 99) is a brand name, it should be substituted with a more comprehensible term both in the article text and in tables & figures; Materials & Method: description is redundant in my opinion, especially some details about food acquisition and preparation, and test execution (lines  89-92, 95, 101-103, 106-107); "level of saliva" expression may be substituted with "amount of saliva" for clarity; Figure 1 is not well-organized from a visual point of view. I suggest to rethink it graphically in order to make it stand out; Sentence at lines 133-135 does not belong to the results Table 1: the concept expressed by the small letters in the columns is not very clear to me, maybe it would be better to include ad additional figure to explain; I think it is mandatory to produce a figure showing the concept expressed in paragraph 3.2, lines 168-173; Line 207 the expression "increased somatosensory access" is not clear; Line 208, "pharyngeal corticobulbar": I think you refer to the corticobulbar tract, but the concept is not clearly explained; Lines 211-218: it is not always clear if you are talking about the present sudy or the study by Asselin [19]; Lines 258-272: maybe you should add a brief explanation of the potential effects of sour product on oral health at this point; Lines 275-279 should not be part of the conclusion in my opinion.

Author Response

The whole article must be submitted to a native English speaker for language revision (introduction and discussion especially). Many terms do not sound very clear (as for example the expression "physiological constellation of food oral processing" is a bit too wordy and confusing)

The manuscript is reviewed by a native English speaker. We will resend for clarifications and change accordingly.

lines 68-69 belong to the conclusion and should not be presented in the introduction;

Thank you for pointing this out. The sentence is now deleted.

"Samarin" (line 99) is a brand name, it should be substituted with a more comprehensible term both in the article text and in tables & figures;

This is absolutely right and the word Samarin is changed into sodium bicarbonate throughout the text.

Materials & Method: description is redundant in my opinion, especially some details about food acquisition and preparation, and test execution (lines  89-92, 95, 101-103, 106-107);

Line 89-92 concerns ethical issues and is necessary to run the test.

Line 95: We agree and the sentence is deleted

Line 101-103: The text is needed to clarify the test procedure and make it possible to repeat

Line 106-107: The text is needed to clarify the test procedure and make it possible to repeat under the same circumstances.

 "level of saliva" expression may be substituted with "amount of saliva" for clarity;

We have changed from level of saliva into amount of saliva

Figure 1 is not well-organized from a visual point of view. I suggest to rethink it graphically in order to make it stand out;

We have updated Figure 1 graphically into a new version

Sentence at lines 133-135 does not belong to the results

The sentence is rephrased

Table 1: the concept expressed by the small letters in the columns is not very clear to me, maybe it would be better to include ad additional figure to explain;

This is a common and accepted way of describing statistically calculated significant differences. The editor should decide whether further explanations are needed.

 I think it is mandatory to produce a figure showing the concept expressed in paragraph 3.2, lines 168-173;

That would be to double report results already given in Table 1. That is normally not allowed in research publications. Again, we leave to the editor to decide

 Line 207 the expression "increased somatosensory access" is not clear;

The somatosensory system is a system of sensory neurons and neural pathways that responds to changes at the surface or inside the body.

Line 208, "pharyngeal corticobulbar": I think you refer to the corticobulbar tract, but the concept is not clearly explained;

Thank you, yes it should be cortibulbar tract. This is corrected. And an explanation is given.

Lines 211-218: it is not always clear if you are talking about the present sudy or the study by Asselin [19];

The text is now clarified.

 Lines 258-272: maybe you should add a brief explanation of the potential effects of sour product on oral health at this point;

One sentence is rewritten.

Lines 275-279 should not be part of the conclusion in my opinion.

This is a part of the conclusion

Reviewer 2 Report

As typically defined, dysphagia refers to a difficulty in swallowing, usually caused by nerve or muscle problems, dysphagia can be painful and is more common in older people. Dysphagia may affect the eating experience and quality of life by causing malnutrition as a result of inadequate nutritional intake.

In this manuscript, Dr. Wendin, K.and co-workers try to investigate different sour foods and carbonated drinks in terms of their efficiency in increasing saliva secretion and triggering swallowing function in relation to foods ingested after the carbonated and/or sour foods. Study results confirmed that sourness and carbonation can manipulate the signal for secretory events and swallowing. In addition, they concluded that there was no order effect, and thereby no correlation between the pH value and the order of ingestion of each test food.

The research study is well designed and appropriately structured. The conclusions are clearly presented. However, I do have a couple of comments that should be addressed and a minor revision is needed.

Particular comments:

Page 1, line 7, use the full name of Dep. Page 8, in the reference part, all the doi information provided are not necessary and can be deleted. Page 9, line 312, the page numbers for Ref. 10 are incorrect. Page 9, line 319, delete “pp”. Some important references are recommended to add to the manuscript. a) Logemann, J.A., Pauloski, B.R., Colangelo, L., Lazarus, C., Fujiu, M. and Kahrilas, P.J., 1995. Effects of a sour bolus on oropharyngeal swallowing measures in patients with neurogenic dysphagia. Journal of Speech, Language, and Hearing Research, 38(3), 556-563. b) Sdravou, K., Walshe, M. and Dagdilelis, L., 2012. Effects of carbonated liquids on oropharyngeal swallowing measures in people with neurogenic dysphagia. Dysphagia, 27(2), 240-250.

Author Response

As typically defined, dysphagia refers to a difficulty in swallowing, usually caused by nerve or muscle problems, dysphagia can be painful and is more common in older people. Dysphagia may affect the eating experience and quality of life by causing malnutrition as a result of inadequate nutritional intake.

In this manuscript, Dr. Wendin, K.and co-workers try to investigate different sour foods and carbonated drinks in terms of their efficiency in increasing saliva secretion and triggering swallowing function in relation to foods ingested after the carbonated and/or sour foods. Study results confirmed that sourness and carbonation can manipulate the signal for secretory events and swallowing. In addition, they concluded that there was no order effect, and thereby no correlation between the pH value and the order of ingestion of each test food.

The research study is well designed and appropriately structured. The conclusions are clearly presented. However, I do have a couple of comments that should be addressed and a minor revision is needed.

Particular comments:

Page 1, line 7, use the full name of Dep.

This is corrected

Page 8, in the reference part, all the doi information provided are not necessary and can be deleted.

The doi is common in most reference lists. The editor may decide whether the information should be deleted

Page 9, line 312, the page numbers for Ref. 10 are incorrect.

Corrected!

Page 9, line 319, delete “pp”.

Corrected

Some important references are recommended to add to the manuscript. a) Logemann, J.A., Pauloski, B.R., Colangelo, L., Lazarus, C., Fujiu, M. and Kahrilas, P.J., 1995. Effects of a sour bolus on oropharyngeal swallowing measures in patients with neurogenic dysphagia. Journal of Speech, Language, and Hearing Research, 38(3), 556-563. b) Sdravou, K., Walshe, M. and Dagdilelis, L., 2012. Effects of carbonated liquids on oropharyngeal swallowing measures in people with neurogenic dysphagia. Dysphagia, 27(2), 240-250.

Thank you. The Dysphagia-paper is added.

Round 2

Reviewer 1 Report

First of all, I want to congratulate for the work done on the manuscript. The paper is now much more easy to read and understand, and I think it is much more likely to catch the reader's interest. So, good job!

Here, few more things I've pointed out for you:

reference 15 (lines 76 and 225): are you sure you are citing the right paper? I like Figure's 1 new design, my only concern is that when I printed out the paper the text in the figure was almost illegible. Could you provide the same figure with higher quality? Figure 3: shouldn't it read "Test food compared to baseline" as explained in the text (156) I still think a figure explaining easiness of swallowing a cracker after the test foods (paragraph 3.2) is crucial. I think this is one of the most important results of the study, and it is also stated in the conclusion. Such figure would serve as to make a concept contained in Table 1 even more evident and graphically more accessible, just like Figure 2 and Figure 3 do.  Lines 185-187 are a bit confusing and might seem to contradict the results of the study. I have understood that it refers to the fact that no order effect was identified, but it should be explained in a different way in order not to mislead the reader. ((Suggestion: maybe you can state it as you wrote in paragraph 2.4, lines 131-132? "correlation between saliva before tasting the test food and ease of swallowing the cracker after the test food")) reference [16] should be put after the sentence that finishes at line 191. line 191: decrease should be substituted with INCREASE ! Paragraph starting at line 217: I am not sure reference [20] matches what is written in text (it was hard to find the reference on the internet). Additionally, I am not sure it can relate to your study, but again, maybe it is because I couldn't find the right publication (https://digitalcommons.unl.edu/cgi/viewcontent.cgi?article=1136&context=ucareresearch) line 243-245: the three products can manipulate saliva secretion (some positively, and sourdough bread negatively even if not significantly as shown in Figure 3). What they stand out negatively for is the fact they do not make swallowing of a neutral food easier afterwards (last column in Table 1). I think this is what should be outlined at this point, considering what comes after in the test. ((otherwise you might say that all test foods except citrus juice, tomatoes and grapefruit did not significantly alter saliva secretion. Or that apples, mango sorbet and sourdough bread decreased the amount of perceived saliva after injection, even if not significantly)) line 249 is not clearly comprehensible to me. Particularly, I don't know where that 80 percent comes from and I think it makes it more confusing. I would do some further adjustment on concepts expressed at paragraph starting at line 263. (for example I would move sentences from line 266 to line 270 and put them after sentence at line 263, and adjust some words in order to make the text flow logically)

Author Response

2nd review. Answer to reviewer in red.

First of all, I want to congratulate for the work done on the manuscript. The paper is now much more easy to read and understand, and I think it is much more likely to catch the reader's interest. So, good job!

Thank you!

Here, few more things I've pointed out for you:

reference 15 (lines 76 and 225): are you sure you are citing the right paper?

Thank you for finding this! The references is now changed

I like Figure's 1 new design, my only concern is that when I printed out the paper the text in the figure was almost illegible. Could you provide the same figure with higher quality?

We send the original to the editor

Figure 3: shouldn't it read "Test food compared to baseline" as explained in the text (156)

The figure text is changed to “baseline cracker”

I still think a figure explaining easiness of swallowing a cracker after the test foods (paragraph 3.2) is crucial. I think this is one of the most important results of the study, and it is also stated in the conclusion. Such figure would serve as to make a concept contained in Table 1 even more evident and graphically more accessible, just like Figure 2 and Figure 3 do. 

We should not report the same data twice. Agree that a figure would be good, but adding a figure and take out the table implies that we will lose information given in the table.

Lines 185-187 are a bit confusing and might seem to contradict the results of the study. I have understood that it refers to the fact that no order effect was identified, but it should be explained in a different way in order not to mislead the reader. ((Suggestion: maybe you can state it as you wrote in paragraph 2.4, lines 131-132? "correlation between saliva before tasting the test food and ease of swallowing the cracker after the test food"))

The text is changed to avoid misunderstandings

reference [16] should be put after the sentence that finishes at line 191.

Done!

Line 191: decrease should be substituted with INCREASE !

Done!

Paragraph starting at line 217: I am not sure reference [20] matches what is written in text (it was hard to find the reference on the internet). Additionally, I am not sure it can relate to your study, but again, maybe it is because I couldn't find the right publication (https://digitalcommons.unl.edu/cgi/viewcontent.cgi?article=1136&context=ucareresearch)

No changes here

line 243-245: the three products can manipulate saliva secretion (some positively, and sourdough bread negatively even if not significantly as shown in Figure 3). What they stand out negatively for is the fact they do not make swallowing of a neutral food easier afterwards (last column in Table 1). I think this is what should be outlined at this point, considering what comes after in the test. ((otherwise you might say that all test foods except citrus juice, tomatoes and grapefruit did not significantly alter saliva secretion. Or that apples, mango sorbet and sourdough bread decreased the amount of perceived saliva after injection, even if not significantly))

We have added to the text that the reader should take look in Table 1.

line 249 is not clearly comprehensible to me. Particularly, I don't know where that 80 percent comes from and I think it makes it more confusing.

The text is changed and the 80% (from correlation calculations) is deleted

I would do some further adjustment on concepts expressed at paragraph starting at line 263. (for example I would move sentences from line 266 to line 270 and put them after sentence at line 263, and adjust some words in order to make the text flow logically)

Thank you, we have changed the text according to this suggestion.
